# Designed for simplicity, used for complexity: The systemic pressures shaping walk-in clinic practices and outcomes

Braeden A. Terpou[1], Lauren Lapointe-Shaw[2,3,4,5,6], Ruoxi Wang[1], Danielle Martin[3,5,7,8], Mina Tadrous[5,6,9,10], Sacha Bhatia[3,6,11], Jennifer Shuldiner[5], Simon Berthelot[12], Niels Thakkar[13], Kerry McBrien[14,15], Bahram Rahman[16], Aisha Lofters[5,6,7,8], J. Michael Paterson[3,6,17], Rita McCracken[18], Christine Salahub[19], Tara Kiran[3,6,7,20], Noah M Ivers[3,5,6,7,8], Laura Desveaux[1,3,5]*

1 Institute for Better Health, Trillium Health Partners, Mississauga, Ontario, Canada, 2 Division of General Internal Medicine and Geriatrics, University Health Network and Sinai Health System, Toronto, Ontario, Canada, 3 Institute of Health Policy, Management and Evaluation, University of Toronto, Toronto, Ontario, Canada, 4 Department of Medicine, University of Toronto, Toronto, Ontario, Canada, 5 Women's College Institute for Health System Solutions and Virtual Care, Women's College Hospital, Toronto, Ontario, Canada, 6 Institute for Clinical Evaluative Sciences, Toronto, Ontario, Canada, 7 Department of Family and Community Medicine, University of Toronto, Toronto, Ontario, Canada, 8 Department of Family Medicine, Women's College Hospital, Toronto, Ontario, Canada, 9 Women's College Research Institute, Women's College Hospital, Toronto, Ontario, Canada, 10 Leslie Dan Faculty of Pharmacy, University of Toronto, Toronto, Ontario, Canada, 11 Peter Munk Cardiac Centre, University Health Network, Toronto, Ontario, Canada, 12 Département de Médecine de Famille et de Médecine D'urgence, Université Laval, Laval, Quebec, Canada, 13 College of Nurses of Ontario, Toronto, Ontario, Canada, 14 Department of Family Medicine, University of Calgary, Calgary, Alberta, Canada, 15 Department of Community Health Sciences, University of Calgary, Calgary, Alberta, Canada, 16 Department of Health Research Methods, Evidence, and Impact, McMaster University, Hamilton, Ontario, Canada, 17 Department of Family Medicine, McMaster University, Hamilton, Ontario, Canada, 18 Department of Family Practice, University of British Columbia, Vancouver, British Columbia, Canada, 19 Toronto General Hospital Research Institute, University Health Network, Toronto, Ontario, Canada, 20 Department of Family and Community Medicine, St. Michael's Hospital, Toronto, Ontario, Canada

* Laura.Desveaux@thp.ca

## Abstract

Walk-in clinics (WICs), appreciated for their accessibility and convenience, have become an increasingly popular healthcare option in Ontario for patients with and without primary care enrolment. Despite their utility, WICs face criticism for delivering lower-quality care compared to comprehensive, enrolment-based primary care models. Critics argue that WICs contribute to system inefficiencies and encourage practice patterns misaligned with population health goals. This study explored physician perspectives on two key outcomes often associated with low-quality care in WICs: repeat primary care visits and potentially inappropriate antibiotic prescribing. Using a qualitative descriptive approach, semi-structured interviews were conducted with Ontario-based family physicians (N = 19) who had experience practicing in both WICs and enrolment-based primary care. The findings highlight systemic challenges, including limited access to enrolment-based primary care and increasing healthcare demands, which have pushed WICs beyond their intended role. This misalignment

**Data availability statement:** All relevant data are within the paper and its Supporting Information files. Excerpts are provided within the body of the paper to highlight exemplary quotes. The minimum data set from which these quotes were pulled can be found in the Supporting Information. All cautions were taken to remove any potentially identifiable information from this attachment.

**Funding:** This work was supported by the Canadian Institutes of Health Research (CIHR) (Grant No. 175285). The funders had no role in study design, data collection and analysis, decision to publish, or preparation of the manuscript.

**Competing interests:** The authors have declared that no competing interests exist.

has created tensions between their structure and purpose, resulting in visits that participants described as more transactional than those in primary care. These constraints—rooted in a lack of informational and relational continuity—often limited participants' ability to provide in-depth engagement or follow-up care. Repeat visits were frequently linked to efforts to ensure continuity for complex or chronic conditions. Similarly, participants acknowledged the reality of potentially inappropriate antibiotic prescribing, attributing it to the high patient volume, desire to satisfy patient expectations, and a tendency to "err on the side of caution" when the nature of the illness is in question. The findings underscore how health system pressures and well-intended policies, such as Ontario's primary care access bonus, can produce unintended consequences, including inequities in access and difficulties with care coordination across settings. Addressing these challenges requires reforms to better integrate WICs with the primary care system, alongside tailored training to support physician decision-making in episodic care contexts.

## Background

Approximately 30% of Ontario residents—nearly 5 million people—visit a walk-in clinic (WIC) each year [1–3]. Among these patients, 94% are already enrolled with a primary care practice [4]. People are drawn to WICs for their convenience [5–9] and timely access [3,4] often using them for conditions that their primary care physician could have addressed [4]. As access pressures intensify in primary care, WIC utilization is expected to rise [10,11], raising questions about whether and how these clinics are evolving to meet this demand and what effects this will have on broader healthcare utilization [10–13].

In Ontario, research on the integration of WICs is limited, with current discussions often anchored in indirect evidence or findings from other countries. This lack of robust, localized data has fueled debates shaped by varying perspectives rather than comprehensive empirical analysis. Proponents of WICs generally argue that they reduce emergency department pressures [14–18] and offer timely access when primary care physicians are unavailable [6,19]. Critics counter that WICs provide lower-quality care compared to comprehensive, enrolment-based primary care models [5–8,20,21–24], citing system inefficiencies, as evidenced by repeat primary care visits, and population health risks, including potentially unnecessary antibiotic prescribing, as key outcomes of concern [8,12,21,22,25,26].

Canadian estimates suggest that approximately 20% of WIC patients follow up with their primary care physician for the same issue within 1–7 days [21], leading to greater healthcare costs [23,24] and complicating care coordination between the two settings [5–9]. However, administrative healthcare data cannot discern whether patients independently sought a second opinion or were intentionally referred back to primary care—and if so, for what reasons [27,28]. Inappropriate

antibiotic prescribing has been another focus of criticism since the early 2000s, when studies first highlighted its prevalence in WICs [29–31]. Approximately 37% of respiratory-related visits to WICs result in an antibiotic prescription, with 8.8% involving upper respiratory infections—conditions for which antibiotics are not generally recommended [32–34]. This pattern contributes to the growing threat of antimicrobial resistance [35,36] and increases the risk of avoidable patient harms [37–39].

In Ontario, Canada's most populous province, the long-standing debate over WIC integration within the healthcare system has intensified, fueled by declining primary care access [2–4] and the rise of virtual WICs [40,41]. Interestingly, nearly 40% of the physicians who work in WICs also practice within enrolment-based primary care models [10]. These dual-role physicians are uniquely positioned to compare and contrast how episodic and continuity-based care models impact healthcare quality and system efficiency, though their voices remain largely absent.

This study explores characteristics of WIC settings that influence repeat primary care visits and potentially inappropriate antibiotic prescribing from the perspectives of physicians practicing in both WICs and enrolment-based primary care models.

## Materials and methods

### Study design

We conducted semi-structured interviews with a cohort of physicians practicing in both WICs and enrolment-based primary care practices in Ontario, Canada. Our primary aim was to investigate the factors that influence repeat primary care visits and potentially inappropriate antibiotic prescribing. This study is part of a larger program of work undertaken to understand whether and how care models can be improved to better meet the care needs of individuals in Ontario [10,11,42].

This study was reviewed and approved by Trillium Health Partners' Research Ethics Board (ID #1091). All participants provided informed consent prior to their participation. Verbal consent was obtained before the interview began and was witnessed by the interviewer. The interviewer used a Verbal Consent Checklist, reading each statement to the participant and confirming their agreement to all included items.

Data collection took place between May 30 and December 6, 2022.

Tong, Sainsbury, and Craig's Consolidated Criteria for Reporting Qualitative Studies (COREQ) [43], a 32-item checklist, was used to ensure explicit and comprehensive reporting of qualitative study procedures.

### Setting

In Ontario, home to over 14.5 million residents, public single-payer health insurance covers emergency department visits, hospital admissions, and medically necessary physician visits without premiums or co-payments. The majority of Ontarians are formally enrolled with a primary care physician, who serves as the first point of contact within the healthcare system [44]. Ontario has around 14,500 physicians practicing within primary care enrolment models [2,3]. These physicians receive capitation payments, based on their roster of patients, comprising between 15% and 80% of their total incomes and are required to provide after-hours access options for patients. An access-related bonus contributes additional income that is reduced when enrolled patients visit outside clinics in-person or virtually [40,45]. This disincentive is a provincial policy that applies specifically to visits coded as primary care, such as WIC visits and some urgent care visits. WICs and urgent care centres in Ontario operate independent of comprehensive primary care settings and as such, do not have access to comprehensive electronic medical records. Both are often staffed by family physicians, some of whom have emergency medicine training. Approximately 10–15% of the population are not enrolled with primary care physicians and rely on fee-for-service physicians, including those in WICs, or emergency departments for episodic and primary care services [3,4,46].

### Recruitment

We employed a purposive sampling strategy to recruit participants, focusing on physicians who were practicing both in in-person or virtual WICs and enrolment-based primary care models in Ontario.

Recruitment was conducted through three parallel channels. First, a list of WICs was compiled using group billing numbers and business names from the Canadian Physician Database and the Ontario Ministry of Health. A *Recruitment Letter* was sent via fax or email to all 472 clinics, asking for their support in distributing an *Invitation Letter* and *Letter of Information* to physicians practicing in their clinics. Second, the research team disseminated the *Invitation Letter* and *Letter of Information* through their professional email networks. Lastly, study details and an invitation to participate were shared on social media platforms, including Twitter, LinkedIn, and Facebook, by the research team members.

To ensure a balanced representation across gender and years in practice, interested participants were asked to provide demographic information. No participants who expressed interest refused to participate after scheduling interviews.

### Data collection

A semi-structured interview guide was developed and piloted with physicians trained in family medicine. Participants were asked to compare their care provider experiences in WICs and enrolment-based primary care practices, and their perceptions of the factors that contribute to these experiences. This study specifically focuses on participants' beliefs about the patient, physician, and contextual factors that influence repeat primary care visits and potentially inappropriate antibiotic prescribing. Questions related to these two indicators comprised of approximately one-third of the interview guide.

Interviews were conducted virtually over Zoom by a Research Associate trained in qualitative interviewing who had no prior relationship with participants. Audio recordings were transcribed and de-identified by an independent third party.

### Data analysis

The transcripts were independently coded by two researchers (RW, BT) using MAXQDA, a qualitative and mixed-methods data analysis software. The analysis was guided by the principles of qualitative description [47,48], employing a primarily deductive approach to map the data to the two outcomes of interest: repeat primary care visits and potentially inappropriate antibiotic prescribing. Deductive codes were developed to identify the drivers underlying these outcomes, structured into two levels. The first level categorized the broader domains of influence, such as system-level pressures, practice model-related pressures, and physician-level pressures. The second level detailed the specific nature of these influences, including, for example, patient needs and access challenges at the system level, financial and demand pressures at the practice model level, and personal beliefs and motivations at the individual level.

After all transcripts were coded deductively, the two researchers collaboratively identified inductive themes that emerged across the deductive codes. Rather than focusing on themes within each outcome, the analysis sought to uncover patterns and relationships across highly co-occurring deductive codes spanning the system, practice model, and individual levels. These inductive themes captured points of intersection where multi-level factors converged to shape the outcomes of interest. Through an iterative process, several themes were developed, and the transcripts were revisited to refine and consolidate the themes, ensuring they accurately reflected the data. Thematic saturation was reached after 15–17 interviews, with an additional 2–4 interviews conducted to confirm that no new insights were identified.

### Results

Nineteen physicians were interviewed, with interviews lasting between 33 and 67 minutes.

The majority of participants identified as female (63%) and the median age was 35 years (see Table 1). Most participants (84%) completed their medical training in Canada and nearly half (47%) were within five years of graduation. Most (95%) practiced in large urban settings, with only one participant based in a smaller community.

**Table 1. Characteristics and demographics of study participants.**

| Demographics | Participants (N = 19) |
|---|---|
| Gender | |
| Male | 7 (37%) |
| Female | 12 (63%) |
| Age | |
| Median [Min, Max] | 35 [29, 57] |
| Medical Training | |
| Canada | 16 (84%) |
| Outside Canada | 3 (16%) |
| Years after Graduating | |
| 0-5 years | 9 (47%) |
| 6-10 years | 6 (32%) |
| 11-20 years | 4 (21%) |
| Average Number of Enrolment-Based Primary Care Hours per Week | |
| 0-9 hours | 6 (32%) |
| 10-19 hours | 3 (16%) |
| 20-29 hours | 8 (42%) |
| 30-39 hours | 2 (10%) |
| Average Number of In-Person WIC Hours per Week | |
| 0-9 hours | 13 (68%) |
| 10-19 hours | 3 (16%) |
| 20-29 hours | 1 (5%) |
| 30-39 hours | 2 (10%) |
| Average Number of Virtual WIC Hours per Week | |
| 0-9 hours | 11 (58%) |
| 10-19 hours | 6 (32%) |
| 20-29 hours | 1 (5%) |
| 30-39 hours | 1 (5%) |

Our findings are structured around three key themes: the pressures imposed on the WIC model by system-level demands, the misalignment of the model as currently designed to effectively address these demands, and the impacts of this misalignment on physician practice, including the tradeoffs they experience that influence repeat family physician visits and antibiotic prescribing.

### Current system demands create strain within the WIC model

The accessibility of WICs emerged as a key driver of utilization, where patients often prioritize the immediacy of care over waiting for an appointment with their primary care physician. The potential impact of these trends on continuity of care was a concern. Despite this concern, some participants, driven by an understanding of the integral role WICs play in meeting patients' immediate needs, opted to work in these settings to support these growing demands.

Participants suggested that the widespread availability of WICs, virtual care, and on-demand services may be inadvertently fueling supply-induced demand by influencing how patients prioritize and pursue care, with same-day access now widely expected by many.

*"It's also the accessibility, right? Like I mentioned, there's so many telemedicine services; a patient could potentially book with all of them, just so that they can be seen now. So, there is kind of a higher demand factor, immediacy of care. Right now, people have higher expectations, right? Why would I wait two hours when I can see someone in ten minutes?" P9 (Female, Early Career (0–5 years))*

Participants described that health anxiety appeared to be more pronounced among patients visiting WICs compared to patients they see in enrolment-based primary care settings. While some of this anxiety could be explained by the acuity of the issues that prompted WIC visits, participants believed it was exacerbated by the intersection of access barriers and the increasing socialization of 'on demand' care.

*"When you're dealing with patients who likely do not know you, now they're not just scared because they have a new health condition, they're scared because they feel like they have no doctor to take care of them. There is a comfort in knowing that you have a family doctor, and patients who do not [have one] bring this discomfort into the visit with them." P15 (Female, Early Career (0–5 years))*

The growing volumes of patients in WICs was complicated by the increasing complexity of cases. Chronic and complex health issues traditionally managed within primary care settings are now appearing with greater frequency at WICs. While WICs serve as a critical safety net for patients without primary care access, participants were troubled by the number of enrolled patients who chose WICs for such issues. They identified patients' perceived urgency and difficulties in accessing primary care as key drivers of this behavior, while also recognizing the need to close the continuity loop with the primary care provider.

### Efforts to address system demands expose misalignments within the WIC model

Participants described the WIC model as inherently high-volume and efficiency-focused, operating primarily under a fee-for-service structure. This design places pressure on physicians to maximize patient throughput, not only to increase individual earnings but also to support the clinic's financial viability amid rising operational costs.

*"If I know I'm only going to get paid $35 for a visit and the patient has needs that would take $400 of my time, I'm going to have to try and navigate that. So, there's economic factors, for sure, that affect decision making, right? There's so many patients we have to see a day to make sure we pay our nurses, staff, EMR and College fees, and the list goes on, ensure we have malpractice insurance. So, you know, if you don't see 15 to 20 people a day, you're already going to be losing money." P11 (Male, Mid-Career (6–10 years))*

Additionally, participants noted that the WIC model's focus on efficiency aligns with the needs and expectations of patients seeking immediate care, creating an environment where WIC physicians must prioritize quick assessments and interventions to manage patient flow and minimize wait times.

*"With the way we're remunerated, it's challenging to make a lot of changes for patients in one visit. It kind of has to be fast. And I can tell you that the demand for services, at least at this WIC, outpaces its supply. So, we have to turn people away every day because there's not enough doctor hours to serve them." P3 (Female, Mid-Career (6–10 years))*

This was complicated by the lack of access to comprehensive medical records—including test results, imaging, or existing care plans—creating significant challenges in WIC practice. One participant likened the experience to "working from scratch" (P14), while others noted that the absence of a detailed health history made them hesitant to alter existing

care plans. This lack of visibility was compounded by the absence of a therapeutic relationship, which participants described as a central to trust building and managing patients' long-term health needs. In primary care, the familiarity cultivated over repeated visits was referred to by some as "social capital" (P6), enabling physicians to navigate challenging conversations with patients. Conversely, when practicing in WIC settings, participants reported having to rely more on patients' self-reported symptoms.

> *"Say for example, a patient comes to you and says I'm having certain pain symptoms that I think are typical of a bladder infection, you as a WIC physician have to kind of take their word for it, because you don't know them, you don't know what their typical bladder infection looks like, so you are more likely to treat that. And because I'm not going to have a subsequent follow-up relationship with [them], I don't have the option to say, 'Listen, I don't think you need antibiotics. Let's give this sickness 24 to 48 hours. If I'm wrong and you're not getting better, tell my secretary, and I will send them over'."* P16 (Female, Early Career (0–5 years))

The episodic nature of WIC visits, combined with the reliance on patients' self-reported symptoms, often made these encounters feel less relational and more transactional. This dynamic increased the pressure to satisfy patient expectations in the WIC setting as compared to their experiences in primary care.

Concerns over patient feedback compounded these social pressures, with participants describing how platforms like RateMyMD or formal complaints through regulatory bodies such as the College of Physicians and Surgeons of Ontario could have serious professional repercussions. Unlike in primary care, where longer appointments and long-term relationships provide opportunities to navigate dissatisfaction and build trust over time, WIC physicians are tasked with achieving patient satisfaction within a single encounter. This consideration can influence clinical decision-making, particularly in scenarios where patient expectations conflict with medical guidelines or best practices.

> *"There's a customer service aspect to [WIC] care, as opposed to the trust and acknowledgment of physician expertise that's more common with primary care patients.*
>
> *I feel there's a significant push across various online platforms and services to encourage people to obtain medications—such as antibiotics or anti-inflammatory drugs—even when they may not be necessary. A classic example is Ozempic, where pharmaceutical companies promote these treatments and direct patients to WICs to obtain them. This approach often lacks proper planning or consideration of the potential consequences of overmedication.*
>
> *I'm not explaining it right but there's an anxiety I know I feel about patients feeling as if they don't get what they want and because of that, there'll be retribution towards you. Whether it's a frivolous complaint or online reviews, when you're actually doing the optimal care, it's often thought to be poor care paradoxically. That's the concern that I have and that keeps me up at night, the fact that the expertise is not appreciated, and it's misunderstood."* P19 (Male, Mid-Career (6–10 years))

### These misalignments result in trade-offs between cost-efficiency, accessibility, and care quality

**Repeat visits.** While participants noted that WIC physicians have the skills to address the growing needs of chronic and complex patients, short appointment times and the reliance on patients' self-reported health histories led many to defer altering medication and treatment plans, if managed by other primary care physicians. Participants noted that patients who are dissatisfied with the depth of care often follow-up with their primary care physician to resolve lingering issues. However, because the patient visited a WIC before consulting them, the primary care physician will receive reduced remuneration due to the outside use penalty triggered by the patient's visit to the WIC.

When WIC physicians did adjust a patient's care plan—a practice several participants acknowledged doing—they strongly preferred that the patient's primary care physician review any management changes, given their familiarity with the patient's health history. For more urgent cases, including complex and evolving issues such as concussions, participants emphasized that follow-up with the primary care physician was not only advisable but clinically necessary.

*"It depends on the nature of the [patient] visit. There are some conditions that evolve over time, like a concussion, and cannot be assessed when the injury or the incident first happens. For those patients, I would suggest follow up, even if it's not required, just so that they have some kind of accountability." P17 (Female, Late Career (11–20 years))*

Though participants acknowledged their role in repeat visits, they collectively underscored that patient preferences for second opinions influence this outcome. The accessibility of WICs ultimately enables patients to seek alternative perspectives, and participants found that patients did so particularly when the treatment or advice provided—either by the WIC physician or their primary care physician—failed to meet their expectations. While some participants found this behaviour frustrating—particularly from their perspective as primary care physicians striving to improve accessibility—others were more understanding, or even supportive, of a patients' right to seek reassurance.

*"I think there's a culture that's almost too anti-second opinion in Canada. I think places like Israel have a better approach, where almost everything gets a second opinion. […] Because here, [as primary care physicians], we're basically saying, if you go to another clinic, I'm firing you. And I think that leads to a lot of dissatisfaction, distrust, and a breakdown in patient–physician relationships, because it's almost like a veiled threat, if not an unveiled one." P19 (Male, Mid-Career (6–10 years))*

Overall, participants reflected that repeat visits, often seen by the healthcare system as inefficiencies or indicators of low-quality care, are frequently driven by the shared goal of patients and providers to maintain continuity, encourage a strong patient experience, and deliver high-quality care within a fragmented system.

**Inappropriate antibiotic prescribing.** Participants felt that the WIC focus on high volumes and efficiency left them poorly equipped to address situations where their clinical judgment differed from the expectations of patients. This was particularly true for antibiotic prescribing, with participants describing that many patients arrived at WICs with deeply held beliefs about the need for such prescriptions. These expectations were common among newcomers to Canada whose beliefs were shaped by cultural norms and experiences in other healthcare systems. Participants reflected that antibiotics are frequently prescribed for respiratory infections in other health systems—often without distinguishing between viral and bacterial causes—leading to a misalignment between patient expectations and the evidence-based prescribing practices physicians were trained to follow.

*"I saw a family of three children last week, all under the age of six. They all had upper respiratory illnesses with conjunctivitis and a mild cough. None of them required antibiotics. The parents were from another country [with much less stringent antibiotic prescribing standards], and so they expected to get them. But, because I was able to see three people at once with the same problem, I was able to take my time, sit down with the family, educate them on, you know, the differences between viral and bacterial infections. I provided a lot of reassurance to them. I was able to make mom and dad feel more comfortable in managing this without any medications, and we didn't have to prescribe antibiotics to three children unnecessarily." P11 (Male, Mid-Career (6–10 years))*

The accessibility and immediacy of WICs were seen as inadvertently reinforcing prescription-seeking behaviours, as patients often viewed WICs as a source of immediate relief. One participant described the challenge of refusing an antibiotic request, explaining that patients would sometimes simply visit another WIC—or even return to the same one

the following day—until their expectations were met. Concerns over patient dissatisfaction complicated these encounters, especially as patients were more likely to express dissatisfaction when a perceived necessary prescription was withheld versus when an unnecessary prescription was provided. The emotional toll of navigating these encounters was evident, with many describing the process as draining and stressful.

> "When I first started, I would try to explain the difference between viral and bacterial infections. That viral [infections] can go away without treatment, whereas bacterial infections require antibiotics. But [after giving my explanation], patients are sometimes like, 'So, are you going to give me antibiotics?' And I feel like I just wasted my time. And so, sometimes it's easier to prescribe so they're happy, even though I know it's not necessary. That's just the reality. When patients come with a strong idea of what they want, it's emotionally exhausting to convince them otherwise." P14 (Female, Early Career (0–5 years))

Participants noted the tension of prescribing antibiotics to address immediate pressures—whether to meet patient expectations or align with the high-efficiency demands of the WIC model. While such decisions temporarily resolved the situation, they often reinforced the very behaviours and expectations participants hoped to discourage, leading to repeated challenges for themselves or other providers. Prescribing decisions exemplified the broader challenge of balancing short-term demands with long-term goals, underscoring the challenges with and complexity of delivering high-quality care in an episodic setting disconnected from other providers.

## Discussion

This study explored factors influencing two outcomes considered to be indicative of low-quality care in WICs: repeat primary care visits and potentially inappropriate antibiotic prescribing. All participants had experience in both WICs and enrolment-based primary care, allowing them to reflect upon their experience in each setting and its implications for practice and quality of care. Using repeat visits and antibiotic prescribing as outcomes of interest, our results highlight broader challenges unique to WICs, stemming from tensions between their intended role and operating structure, and the growing demands for timely and convenient access to care.

Our findings illustrate the paradox that while WICs were introduced as a stop-gap for non-emergency care [5,6,19], the increasing pressures on Ontario's healthcare system are pushing WIC providers to provide services they were not intended to offer. These pressures are influenced by declining access to enrolment-based primary care [1,49–51] and widening inequities in outcomes [52–54]. While WICs remain a critical access point for those without primary care, the frequent use by enrolled patients [10,11,41] suggests the use of WICs has evolved beyond the intended stop-gap to a primary point of access for all [55,56], potentially undermining the primary care system they were meant to complement [4,46,57,58].

Participant experiences highlight how well-intended recent policies, including Ontario's primary care access bonus (designed to encourage after-hours availability in enrolment-based primary care and reduce WIC utilization by enrolled patients [40,45]), can produce unintended consequences when they overlook system-level complexities [44,46,58]. Some primary care physicians, in an effort to maintain eligibility for this bonus, deroster patients who frequently use WICs [59–61]. This practice may exacerbate access challenges, particularly for individuals with complex health needs and higher states of health anxiety, who often turn to WICs for reassurance and timely care when primary care access is limited, or individuals who live in large urban areas, where WIC accessibility is higher [62,63]. Our findings suggest that policies should account for the complexity of access needs, recognizing that advising follow-up as a WIC physician—resulting in repeat visits—often reflects efforts to uphold continuity and comprehensive care, particularly in cases where informational or relational continuity is essential for delivering high-quality, integrated care.

Participants described WIC visits as feeling more transactional [12,64,65], in the absence the relational continuity that serves as the foundation for collaborative decision-making between patient and physician [20,66]. They felt that the WIC model limits the depth of physician–patient engagement, making decision-making more responsive to immediate patient needs and expectations. While this responsiveness can empower patients [65,67,68], participants noted that it often left them feeling less autonomy in comparison to their experiences in enrolment-based primary care models, where ongoing relationships allow for dialogue around a broader, long-term view of patient care. While participants often took the position that meeting expectations for antibiotics led to patient satisfaction, some studies suggest that the interpersonal skills of the WIC physician are a much stronger predictor of satisfaction [68,69], suggesting satisfaction extends beyond meeting immediate needs. This underscores the need for future research to explore how interpersonal skills—such as active listening, empathy, and collaborative communication—can be adapted to the unique constraints of WIC visits. As the social dynamics of these visits differ from those in enrolment-based primary care practice, tailored training and resources may be necessary to assist physicians with improving both patient experiences and adherence to clinical guidelines [70,71].

Although our study focused on identifying shared themes among dual-role WIC physicians, a deeper exploration of provider-specific characteristics—such as demographic factors (age, sex, and experience), as well as their perspectives, beliefs, and interpersonal skills—could offer meaningful insights into how different physicians navigate difficult conversations in high-volume practice settings. Rodrigues et al.'s recent systematic review found that physicians' attitudes, namely complacency and fear, are linked to an increased likelihood of prescribing antibiotics [72]. Investigating whether these tendencies are more prevalent among early-career providers or among physicians with certain communication styles could shed light on how these factors influence prescribing behaviour in high-volume practice settings [72–74]. This knowledge could help develop tailored interventions designed to reinforce physician confidence and equip them with strategies to manage patient expectations effectively [74], similar to work conducted by van Buul et al in nursing homes [75]. While this was beyond our study's scope, it represents an important avenue for future research.

These experiences beg the question as to what motivates physicians who choose to practice in WICs. Our prior work details how many are drawn to the flexibility of the WIC model, which offers adaptable hours, higher compensation, and the opportunity to balance other professional practice pursuits [5,7–9,42]. For some, WICs offer an on-ramp to build their primary care practice, while other view it as an alternative to enrolment-based primary care that allows them to practice medicine without the long-term accountability of a rostered primary care practice [5,7,42].

Physicians practicing in the United States similarly cite financial and administrative pressures as driving forces behind their shift from independent practice to employment-based models [76–79]. Declining reimbursements, rising operational expenses, and the financial toll of the COVID-19 pandemic have made the flexibility and stability of employment-based practice more attractive to many. These organizations are seen as alleviating physicians from the burdens of running a private practice, allowing them to focus more on clinical work [42,76,77,79]. Health systems must simultaneously consider push factors (i.e., the increasing challenges of running a primary care practice) and pull factors (i.e., the flexibility and compensation offered by WICs) when designing strategies to improve access to comprehensive primary care. Addressing these factors may make primary care a more attractive and sustainable pathway for a wider range of physicians [46,62,63], while alleviating WICs of many of the complex and chronic presentations that create tension for the WIC model and the physicians practicing within it.

## Limitations

These findings reflect the perspectives of Ontario-based physicians trained in family medicine, and may not be representative of physicians from other regions. Unlike recent studies that found WIC physicians are typically men in the late stages of their careers [10], our sample was largely composed of early-career women. While our participant sample supported in-depth insights from those actively navigating the landscape in both WICs and comprehensive primary care, future work is needed to understand whether and how these insights resonate with other physician populations, whose

perspectives may differ in meaningful ways. Early-career women physicians may experience unique pressures, and future work should explore variability in experiences related to workload, patient expectations, or role flexibility across different physician demographics. It is unclear whether their perspectives reflect broader trends among WIC providers or if their differing demographics influence their approach to patient demands, particularly regarding inappropriate antibiotic prescriptions. Experienced providers may have developed strategies to manage patient expectations, while newer providers might either feel more confident in asserting best practices or, conversely, be more vulnerable to patient pressure. Secondly, Ontario's healthcare landscape includes a variety of episodic care models, including WICs (characterized by stand-alone facilities that focus on acute, non-emergency care), urgent care centres, and retail clinics operated by healthcare networks or for-profit corporations. Most study participants practiced in WICs and served predominantly urban populations, which limits the applicability of the findings to physicians working in retail clinics or those serving rural areas. Thirdly, as participation was voluntary, our sample may be subject to selection bias; physicians who were dissatisfied with aspects of their practice or who valued the WIC model were likely more inclined to participate. Lastly, response bias may have played a role, with physicians potentially presenting their primary practice setting—whether WIC or primary care—in a more favourable light.

## Conclusion

While WICs offer valuable access to care, our findings suggest the rise in repeat visits and potentially inappropriate antibiotic prescribing are symptoms produced by the current structure of WICs and their ability to respond to health system pressures. As WICs evolve, bridging the gap between their intended role and current demands will be critical to advancing patient-centred care and improving health outcomes at a system level. Reforming policy and reimbursement structures to address the nuances between WICs and the primary care system can facilitate integration between the two, enabling high-quality care to be delivered across care settings without undermining the integrity of the primary care system.

## Supporting information

**S1 File. Interview guide.**
(DOCX)

**S2 File. Supporting quotes.**
(DOCX)

## Acknowledgments

We extend our sincere gratitude to the physicians who generously shared their time, experiences, and insights for this study. Their thoughtful reflections have been invaluable in deepening our understanding of the complexities and challenges providers face when practicing in these different practice settings.

## Author contributions

**Conceptualization:** Braeden A. Terpou, Lauren Lapointe-Shaw, Danielle Martin, Noah M Ivers, Laura Desveaux.

**Data curation:** Laura Desveaux, Braeden A. Terpou.

**Formal analysis:** Braeden A. Terpou, Ruoxi Wang, Laura Desveaux.

**Funding acquisition:** Lauren Lapointe-Shaw, Noah M Ivers, Laura Desveaux.

**Investigation:** Braeden A. Terpou, Lauren Lapointe-Shaw, Laura Desveaux.

**Methodology:** Braeden A. Terpou, Laura Desveaux.

**Resources:** Lauren Lapointe-Shaw.

**Software:** Laura Desveaux.

**Supervision:** Laura Desveaux.

**Validation:** Danielle Martin, Mina Tadrous, Sacha Bhatia, Jennifer Shuldiner, Simon Berthelot, Niels Thakkar, Kerry McBrien, Bahram Rahman, Aisha Lofters, J. Michael Paterson, Rita McCracken, Christine Salahub, Tara Kiran, Noah M Ivers, Laura Desveaux.

**Visualization:** Braeden A. Terpou.

**Writing – original draft:** Braeden A. Terpou, Laura Desveaux.

**Writing – review & editing:** Braeden A. Terpou, Lauren Lapointe-Shaw, Ruoxi Wang, Danielle Martin, Mina Tadrous, Sacha Bhatia, Jennifer Shuldiner, Simon Berthelot, Niels Thakkar, Kerry McBrien, Bahram Rahman, Aisha Lofters, J. Michael Paterson, Rita McCracken, Christine Salahub, Tara Kiran, Noah M Ivers, Laura Desveaux.

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
