## [Decision Letter · Decision Letter 0]

25 Feb 2025

PLOS ONE

Dear Dr. Laura Desveaux,

Thank you for submitting your manuscript to PLOS ONE. After careful consideration, we feel that it has merit but does not fully meet PLOS ONE’s publication criteria as it currently stands. Therefore, we invite you to submit a revised version of the manuscript that addresses the points raised during the review process.

We look forward to receiving your revised manuscript.

Kind regards,

Faten Amer, PhD in Health Sciences

Academic Editor

PLOS ONE

Journal Requirements:

 “Funding was received from the Canadian Institutes of Health Research (CIHR) project grant awarded to Lauren Lapointe-Shaw, Noah Ivers and Laura Desveaux (no. 175285).”        

3. We note that you have indicated that there are restrictions to data sharing for this study. PLOS only allows data to be available upon request if there are legal or ethical restrictions on sharing data publicly. For more information on unacceptable data access restrictions, please see http://journals.plos.org/plosone/s/data-availability#loc-unacceptable-data-access-restrictions .  

Reviewers' comments:

Reviewer's Responses to Questions

**Comments to the Author**

1. Is the manuscript technically sound, and do the data support the conclusions?

Reviewer #1: Yes

2. Has the statistical analysis been performed appropriately and rigorously?

Reviewer #1: N/A

3. Have the authors made all data underlying the findings in their manuscript fully available?

Reviewer #1: No

4. Is the manuscript presented in an intelligible fashion and written in standard English?

Reviewer #1: Yes

Reviewer #1: The purpose of this analysis was to gather the perspectives from providers that work in both walk-in clinics (WIC) and enrollment-based primary care practice. These perspectives could help inform researchers of repeated WIC visits and inappropriate antibiotic prescribing. This was primarily a qualitative study based around semi-structured interviews of 19 providers who were recruited from three parallel channels: WIC clinics, professional email networks, and social media platforms. Interviews were done through video conferencing and ultimately transcribed to pull out themes relevant to the research question in iterative inductive and deductive processes. The responding sample was more female, early career, and urban than typical WIC providers. The key themes of their interviews were WIC accessibility, the patient expectation of immediate care, the increasing complexity of cases, the patient ignorance and demand of antibiotics, and the effect of a patient review system. Based on the analysis, authors concluded that WIC are purposed for tasks outside their original design in response to health system pressures, which highlights the need to reform policy and reimbursement practices.

Overall, I found this study to be well-structured and informative.

I am interested in having a deeper understanding about whether there were any concentration or trend of themes according to the provider's age, sex, work division between WIC/primary care office, and career stage. Since this population seems to differ from the typical WIC provider, are these respondents more or less confident in telling a patient "no" for inappropriate antibiotics? Are seasoned providers telling patients "no", but newer providers bending to pressure? Or are these providers enthusiastic being just out of school and able to assert themselves where veteran providers are worn down by the system and therefore allow patients to walk over them? The possible response bias is an aspect that I feel you should spend just a little more time on in the report. I know you already have addressed it partially in the limitations, but I would dig into it just a bit more.

Finally, I request that you provide additional information in the supplemental data. I would like to see what the interview guide is and also a kind of aggregation of quotes that support the different themes / key messages from across the interviews (i.e., not just the quotes put in the manuscript). This would support that the theme was shared rather than sourced from only one person.

**Do you want your identity to be public for this peer review?** For information about this choice, including consent withdrawal, please see our Privacy Policy

Reviewer #1: **Yes: ** Lauren Parlett

---

## [Author Response · Author response to Decision Letter 1]

14 Apr 2025

Manuscript ID: PONE-D-24-57353

Manuscript Title: Misaligned mandates: The systemic pressures shaping walk-in clinic practices and outcomes

We would like to thank the Academic Editor, Faten Amer, and the reviewers for their thoughtful and constructive feedback to help improve the clarity and impact of the manuscript. An itemized response for each comment is presented below. Please note that references to page and line numbers correspond to the revised version of the manuscript.

Editor’s Comments:

1. Funding Disclose:

Thank you for stating the following financial disclosure:

“Funding was received from the Canadian Institutes of Health Research (CIHR) project grant awarded to Lauren Lapointe-Shaw, Noah Ivers and Laura Desveaux (no. 175285).”

Author’s Comments:

We have revised the Cover Letter to include the statement provided:

2. Data Sharing:

We note that you have indicated that there are restrictions to data sharing for this study. PLOS only allows data to be available upon request if there are legal or ethical restrictions on sharing data publicly.

A.) If there are ethical or legal restrictions on sharing a de-identified data set, please explain them in detail (e.g., data contain potentially identifying or sensitive patient information, data are owned by a third-party organization, etc.) and who has imposed them (e.g., a Research Ethics Committee or Institutional Review Board, etc.). Please also provide contact information for a data access committee, ethics committee, or other institutional body to which data requests may be sent.

B.) If there are no restrictions, please upload the minimal anonymized data set necessary to replicate your study findings to a stable, public repository and provide us with the relevant URLs, DOIs, or accession numbers.

You also have the option of uploading the data as Supporting Information files, but we would recommend depositing data directly to a data repository if possible.

Author’s Comments:

We thank the editor for raising this concern. Adhering to PLOS ONE's Data Availability Guidelines for Qualitative Data, we have included relevant excerpts directly in the paper. However, due to ethical restrictions regarding participant consent for data release, additional data cannot be made publicly available as the data may contain personally identifiable information. This decision aligns with Trillium Health Partners’ Research Ethics Board. For further inquiries, interested parties may contact Trillium Health Partners’ Research Ethics Board (THPREB@thp.ca).

To provide a more comprehensive view of the data, and in response to reviewer comments, we have added additional quotes as a Supplementary File. All quotes have been de-identified and prepared in line with governing REB requirements for sharing qualitative data.

3. References:

Author’s Comments:

We have reviewed the reference list and did not find any errors. If there are specific references that the editor believes require our attention, please let us know.

Reviewer’s Comments:

Reviewer 1:

1. Discussion:

I am interested in having a deeper understanding about whether there were any concentration or trend of themes according to the provider's age, sex, work division between WIC/primary care office, and career stage. Since this population seems to differ from the typical WIC provider, are these respondents more or less confident in telling a patient "no" for inappropriate antibiotics? Are seasoned providers telling patients "no", but newer providers bending to pressure? Or are these providers enthusiastic being just out of school and able to assert themselves where veteran providers are worn down by the system and therefore allow patients to walk over them? The possible response bias is an aspect that I feel you should spend just a little more time on in the report. I know you already have addressed it partially in the limitations, but I would dig into it just a bit more.

Authors’ Comments:

Thank you for your encouragement, comments, and feedback. We have revised the Discussion to include a deeper dive into physician characteristics:

“Our findings align with Rodrigues et al.’s systematic review, which identified complacency and fear as key attitudes linked to increased antibiotic prescribing [76]. Additionally, burnout—another theme raised by our participants—has been shown to influence prescribing behavior. Derricks et al. found that physicians experiencing higher levels of burnout were more likely to prescribe opioids and at higher doses for patients with advanced lung cancer [77]. These findings suggest that external pressures, including emotional and cognitive strain, can shape prescribing decisions across different clinical contexts. Investigating whether there are subgroups of physicians for whom the tendency to prescribe is more prevalent could further illuminate how these factors influence practice patterns in high-volume practice settings [78, 79]. Research from the primary care literature indicates that prescribing behaviors do differ based on physician characteristics. For example, a U.S. study found that female general practitioners (GPs) working in the Veterans Administration were less likely to be in the highest prescribing quartile for benzodiazepines, opioids, PPIs, and antibiotics [80]. Moreover, in a systematic review using an international sample, Baillie et al. found that early-career GPs had lower odds of prescribing antibiotics compared to their more experienced peers (OR 0.23–0.67) [81]. These findings suggest that tailored interventions could be valuable in reinforcing physician confidence and providing them with structured strategies to navigate patient expectations [79]. Although no studies have specifically examined the effectiveness of tailored interventions for prescribing decisions, Joosen et al. demonstrated that occupational physicians significantly improved adherence to mental health guidelines after receiving multiple-session peer training designed to identify and address implementation barriers using a Plan-Do-Check-Act approach [82]. Future research could build on this framework by developing physician personas, similar to the work of Shuldiner et al. [83], to inform the co-design of tailored intervention strategies.” (Page 15, Line 6–28)

We also elaborated further on response bias as a limitation:

“These findings reflect the perspectives of Ontario-based physicians trained in family medicine, and may not be representative of physicians from other regions. Unlike recent studies that found WIC physicians are typically men in the late stages of their careers [10], our sample had a larger proportion of women who were early-career. In particular, a study by Lapointe-Shaw et al found that across the 597 WIC physicians practicing in Ontario in 2019, 67% were male with an average years of experience of 25.5 [10]. By contrast, 63% of our participants were female and 47% were early career (vs. 8.5% reported by Lapointe-Shaw et al. [10]). As the intention of this study was to generate insights into what might be contributing to system outcomes, these insights should not be taken to be representative of all family physicians. Future work should explore whether these perspectives resonate with a larger sample of WIC providers, as well as investing whether and how different physician characteristics influence their approach to patient care. For example, more experienced providers may have developed effective strategies to manage patient expectations, while newer providers may be more vulnerable to patient pressure. Conversely, early career physicians may be more familiar with current evidence and firm in their approach to clinical care. Identifying the beliefs that drive physician decision-making and among which physicians these beliefs are most common is a necessary precursor to developing effective strategies to improve population outcomes.” (Page 16, Line 22 – Page 17, Line 10)

2. Supplemental Materials:

Finally, I request that you provide additional information in the supplemental data. I would like to see what the interview guide is and also a kind of aggregation of quotes that support the different themes / key messages from across the interviews (i.e., not just the quotes put in the manuscript). This would support that the theme was shared rather than sourced from only one person.

Authors’ Comments:

We have added the Discussion Guide and a document containing a broader selection of quotes beyond those featured in the manuscript. These quotes are organized by theme to help readers appreciate the nuance of each theme and the corresponding coverage across participants. We hope these additions address the reviewer’s concerns and further support the validity of our findings.

---

## [Editor Report · Decision Letter 1]

12 May 2025

Dear Dr. Desveaux,

Thank you for submitting your manuscript to PLOS ONE. After careful consideration, we feel that it has merit but does not fully meet PLOS ONE’s publication criteria as it currently stands. Therefore, we invite you to submit a second revision of the manuscript that addresses the points raised during the review process.

We look forward to receiving your revised manuscript.

Kind regards,

Julie Gleason-Comstock

Academic Editor

PLOS ONE

Journal Requirements:

Additional Editor Comments (if provided):

1. Sample size: The sample of 19 physicians is relatively small, with the added consideration of the majority of respondents being early career women. As noted in Discussion and Limitations, this is an important consideration.

2. Ontario, Canada health service delivery: Additional detail regarding the differences and similarities of service delivery in Ontario, as a Province in Canada, and Walk in Clinics (WIC) and Urgent Care would be helpful. For example, it appears both do not have access to comprehensive medical records. However, WIC may be able to provide dental and eye care, whereas only Urgent Care can treat fractures and injuries requiring stitches? Are the majority of WIC staffed by family physicians or do emergency medicine physicians play a role as well?

3. Renumeration: Is it a national or provincial policy regarding the “penalty” triggered for accessing WIC before primary care, or does that also relate to urgent care?

4. “Misaligned Mandates” is an interesting title – it would be helpful to learn more about the actual legislation or mandates.

---

## [Author Response · Author response to Decision Letter 2]

15 May 2025

Response to Reviewers

We would like to thank the editor for their clarifying questions to improve the manuscript. An itemized response for each comment is presented below.

Editor Comments

1. Sample size: The sample of 19 physicians is relatively small, with the added consideration of the majority of respondents being early career women. As noted in Discussion and Limitations, this is an important consideration.

Response: We have added the following to the limitations section to ensure due consideration:

“Unlike recent studies that found WIC physicians are typically men in the late stages of their careers [10], our sample was largely composed of early-career women. While our participant sample supported in-depth insights from those actively navigating the landscape in both WICs and comprehensive primary care, future work is needed to understand whether and how these insights resonate with other physician populations, whose perspectives may differ in meaningful ways. Early-career women physicians may experience unique pressures, and future work should explore variability in experiences related to workload, patient expectations, or role flexibility across different physician demographics.”

2. Ontario, Canada health service delivery: Additional detail regarding the differences and similarities of service delivery in Ontario, as a Province in Canada, and Walk in Clinics (WIC) and Urgent Care would be helpful. For example, it appears both do not have access to comprehensive medical records. However, WIC may be able to provide dental and eye care, whereas only Urgent Care can treat fractures and injuries requiring stitches? Are the majority of WIC staffed by family physicians or do emergency medicine physicians play a role as well?

Response: We have added the following context at the end of the ‘Setting’ section (note that WICs in Ontario do not provide dental or vision care).

“WICs and urgent care centres in Ontario operate independent of comprehensive primary care settings and as such, do not have access to comprehensive electronic medical records. Both are often staffed by family physicians, some of whom have emergency medicine training.”

3. Renumeration: Is it a national or provincial policy regarding the “penalty” triggered for accessing WIC before primary care, or does that also relate to urgent care?

Response: We have added more context to the Setting subsection as outlined below:

“An access-related bonus contributes additional income that is reduced when enrolled patients visit WICs in-person or virtually. This disincentive is a provincial policy that applies specifically to visits coded as primary care, such as WIC visits and some urgent care visits.”

4. “Misaligned Mandates” is an interesting title – it would be helpful to learn more about the actual legislation or mandates.

Response: The initial title refers to implicit expectations rather than formal mandates. To avoid confusion for those who aren’t familiar with the context, we have updated the title to be more explicit: Designed for Simplicity, Used for Complexity: The systemic pressures shaping walk-in clinic practices and outcomes.

---

## [Editor Report · Decision Letter 2]

20 May 2025

Designed for simplicity, used for complexity: The systemic pressures shaping walk-in clinic practices and outcomes

PONE-D-24-57353R2

Dear Dr.. Desveaus:

We’re pleased to inform you that your manuscript has been judged scientifically suitable for publication and will be formally accepted for publication once it meets all outstanding technical requirements.

Kind regards,

Julie Gleason-Comstock, PhD, MCHES

Academic Editor

PLOS ONE

---

## [Editor Report · Acceptance letter]

PONE-D-24-57353R2

PLOS ONE

Dear Dr. Desveaux,

I'm pleased to inform you that your manuscript has been deemed suitable for publication in PLOS ONE. Congratulations! Your manuscript is now being handed over to our production team.

Kind regards,

on behalf of

Professor Julie Gleason-Comstock

Academic Editor

PLOS ONE